# Sedimentary and reservoir characterization of wandering braided river: a case of the Xiashihezi Formation 1 member in Shilijiakhan west zone, Hangjinqi, Northern Ordos Basin

**Hongtao Li** *, **Tao Liu, Qingbin Liu**

Petroleum exploration and Development Research Institute, Sinopec, Beijing, China

* liht.syky@sinopec.com

## Abstract

The Ordos Basin's Hangjinqi Shiligahan west zone Xiashihezi Formation 1 Member gas reservoir exhibits significant exploration and development potential. However, its sedimentation and reservoir characteristics are poorly understood. To address this, geological, seismic, macroscopic, and microscopic methods are combined. The available data includes core observation, thin sections, physical property analysis, mercury injection curve interpretation, seismic attribute predictions, and so on. The results show that the target stratum lithology and sedimentary structure are complex and diverse. They are dominated by gravelly coarse sandstone facies with strong hydrodynamic conditions, intercalated with thin mudstone and siltstone. The vertical meter-scale cycle sedimentation characteristics are distinct, representing a typical shallow braided river deposition. Through the mutual calibration and verification of lithology, logging, and seismic facies, the braided channel and its internal microfacies boundaries were accurately delineated. The reservoir primarily consists of gravel-bearing coarse sandstone, featuring intergranular and intragranular dissolved pores as the primary types of reservoir space. The pore types are predominantly mesopores and small pores, with pore-throat combinations favoring mesopores and medium throats, as well as medium to small pores and fine throats. The reservoirs average porosity and permibility are 7.6% and 0.53mD, respectively. This defines a typical reservoir with low to ultra-low porosity and low to ultra-low permeability. The sedimentary microfacies and their associated hydrodynamic conditions are crucial for the development of the reservoir in the Xiashihezi Formation 1 Member. This leads to variations in reservoir properties and pore structures. Thus, the reservoirs are predominantly located in the channel bar microfacies, clearly demonstrating characteristics of facies-controlled reservoir development.

**Data availability statement:** All relevant data are within the manuscript and its Supporting Information files.

**Funding:** Fund projects: "Research on the Development Potential of Dense and High Water-bearing Gas Reservoirs in Dongsheng Gas Field" (P23133) by the Ministry of Science and Technology of Sinopec and National Major Special Project "Development of Large Oil and Gas Fields and Coalbed Methane" (No. 2016ZX05002-006).

**Competing interests:** The authors declare that they have no known competing financial interests or personal relationships that could have appeared to influence the work reported in this paper.

## Introduction

There are many sets of sandtone bodies developed in Permian in Ordos Basin. Among them, the reservoir sandstone bodies of the Xiashihezi Formation 1 Member are the most developed, which are vertically overlapped and horizontally combined, and have the characteristics of wide coverage spatial distribution [1]. This set of sandstone body reservoir has a controlled reserve of more than 100 billion $m^3$ in the west zone of Shiligahan, Hangjinqi, with great exploration and development potential [1–6], and is one of the main target areas for increasing reserves and production in the future. However, low porosity and permeability and poor physical properties lead to low production degree of the gas reservoir. In order to further explore and develop the tight and low-permeability gas reservoirs in Ordos Basin, it is urgent to understand the sedimentary and reservoir characteristics of such gas reservoirs, clarify the impact of sedimentation on reservoir development, and provide the basis for further macro distribution analysis of reservoirs. Previous studies have shown that the gas reservoir type of the Xiashihezi Formation 1 Member in Shiligahan west zone, Hangjinqi, northern Ordos Basin was dominated by lithologic gas reservoirs [6,7]. Therefore, sedimentary microfacies was one of the important controlling factors for the development of reservoirs and gas reservoirs. However, there ws a great controversy on the understanding of the sedimentary characteristics of the western Shiligahan west zone by predecessors. Some scholars believed that it was alluvial fan sedimentation, while others believed that it was braided river sedimentation [8–10], and the understanding of sedimentation was not unified. In terms of sedimentary microfacies boundary delineation, predecessors used logging data to carry out single well sedimentary microfacies division, and used the difference method between wells to speculate sedimentary microfacies [3]. However, the study area is large, the degree of well control is low, and the deposition of the Xiashihezi Formation 1 Member is complex. The predecessors did not fully use seismic data to depict the microfacies boundary [3,11], and the plane distribution of sedimentary microfacies often has a large error with the actual production, that is, its predictability is poor. Although some scholars have preliminarily determined the distribution of sandstone bodies by using the geophysical arc length attribute, and have inversed the thickness of sandstone bodies in the Xiashihezi Formation 1 Member through geophysics, and compiled the sandstone strata ratio map [12], they have not really effectively depicted the sedimentary microfacies boundary. The sedimentary microfacies in this area change rapidly laterally. Facing the research object with such complex sedimentary laws, it is obviously difficult to accurately describe the plane distribution of sedimentary microfacies by using only limited wells data or sandstone strata ratio data. More importantly, the Xiashihezi Formation 1 Member in the west zone of Shiligahan area has diverse lithological assemblages, compact reservoir structure, complex reservoir space types and strong heterogeneity. The previous analysis of reservoir characteristics in the study area is relatively few, usually only the lithological and physical characteristics are analyzed, and the analysis of reservoir space type and pore structure is basically not carried out [12,13]. Therefore, the understanding of reservoir also has certain

limitations. The pore structure and its combination characteristics can reflect the reservoir and seepage ability to control the occurrence state and seepage characteristics of reservoir fluid, and thus is one of the key contents of tight sandstone reservoir evaluation [14,15].

In summary, there is a need to enhance understanding of sedimentation and reservoir characteristics in this area, along with further research into how sedimentation influences and controls reservoir development, which in turn poses a challenge for exploration and development. Clearly, precisely delineating the favorable sedimentary facies of the gas reservoir, defining the boundaries of the favorable reservoir, and conducting research on fundamental reservoir characteristics such as basic petrology, physical properties, reservoir space types, and pore structure within these boundaries are fundamental to evaluating and predicting oil and gas reservoirs. These efforts are central to quantitative reservoir characterization and are essential prerequisites for understanding the gas reservoir. They provide a basis for precise and detailed description and evaluation of the gas reservoir and are key to its further effective development.

In this study, based on detailed observations of drilling coring and a range of data including geological, logging, and seismic information, the well and seismic data was closely integrated, and multidisciplinary combination research was carried out. Through integrated geological, logging, and seismic research, we investigated reservoir sedimentary characteristics, optimized geophysical sensitive attribute parameters, and precisely delineated microfacies boundaries to ascertain the reservoir's macro distribution. Further analyses, including thin section observations, physical property measurements, and mercury injection curve evaluations, described reservoir lithology and space types, quantified physical properties, analyzed pore structure and other microscopic characteristics, and summarized the impact of sedimentation on reservoir development. The research results not only offer insights for the analysis of similar braided river tight sandstone reservoirs but also serve as a basis for further reservoir studies in this area, thus bearing significant theoretical and practical implications.

## Regional geological background

The Hangjinqi area and its adjacent area are located in the northern Ordos Basin and in the west of Inner Mongolia Autonomous Region and the northwest of Ordos City, in the Ordos Plateau and Hetao Plain. The current tectonic position is located in the junction of Yimeng uplift and Yishan slope in the northern Ordos Basin. This area has been in a high position of the basin structure for a long time and is a favorable direction for natural gas migration. In general, the structure is monoclinic, inclined to the southwest, and a nosing structure with low amplitude trending to the northeast was developed locally. The north side of the Shilijiahan west zone is bordered by the Boerjianghaizi fault, and the west side is adjacent to the Wulanjilinmiao fault. The strike of the two faults is nearly east-west (Fig 1).

In the Late Carboniferous period, the southern area of Otogqi and Ejinhoroqi in the Hangjinqi region deposited the Carboniferous Taiyuan Formation as a result of weathering and erosion of the lower Ordovician carbonate rocks. In the early Permian, the transgression reached a climax, and the sedimentary range gradually expanded to the area of the Boerjianghaizi fault, depositing the Permian Shanxi Formation, the Xiashihezi Formation, the Shangshihezi Formation, and the Shiqianfeng Formation [2]. The Xiashihezi Formation was composed of (sandy) conglomerate, gravelly coarse sandstone, grayish green medium fine sandstone, silty mudstone and grayish green, brown mudstone. The middle and lower part of Xiashihezi Formation 1 Member was mainly composed of (sandy) conglomerate, gravelly lithic coarse sandstone, lithic quartz sandstone, which was an important reservoir development layer. The Shangshihezi Formation and Shiqianfeng Formation mainly deposited fine-grained sedimentary rocks.

## Comprehensive analysis of sedimentary facies and its distribution characteristics

**Sedimentary facies analysis and logging response characteristics.** The detailed core observation revealed that the lithology of the sandstone body of Xiashihezi Formation 1 Member was complex and diverse in the research area.

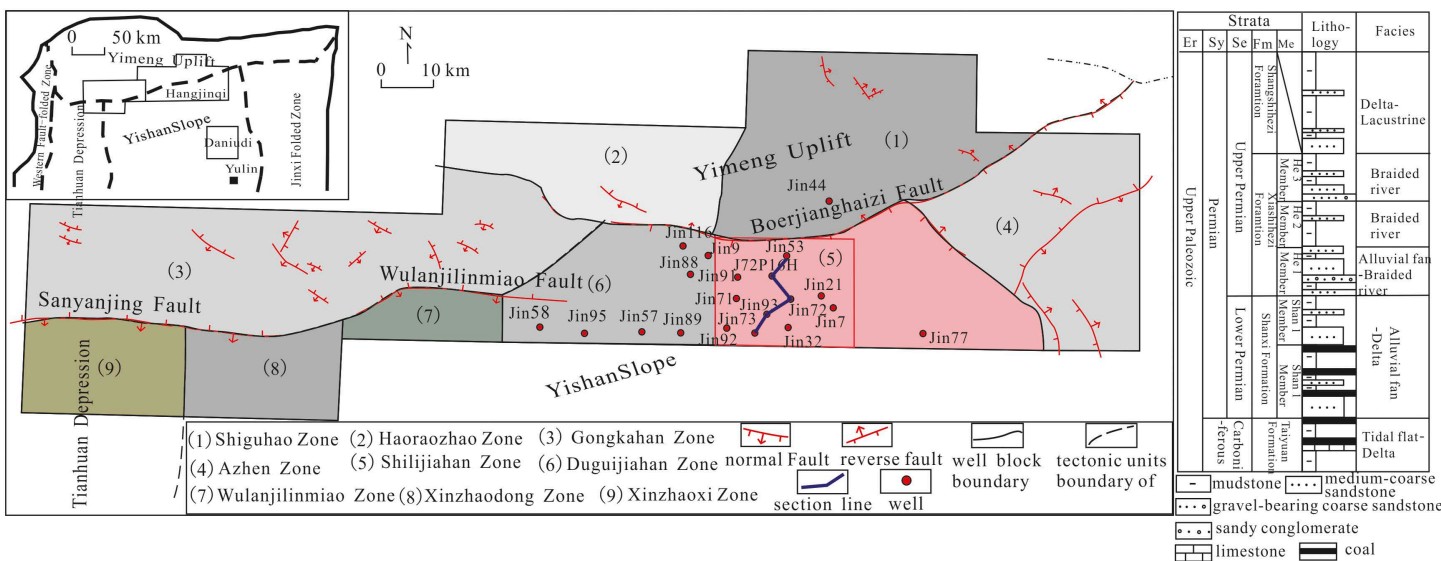

**Fig 1. Map showing tectonic unit division in Ordos Basin and loction of Shilijiahan west zone, Hangjinqi area.**

Variegated (sandy) conglomerate was developed at the bottom of some channel sedimentary sequences, which was mainly composed of medium conglomerate and mostly composed of multi-component lithic conglomerate and mud gravel was rare. The gravel was poorly sorted but relatively well rounded. Gravel was slightly imbricated in some places, and calcite cementation and filling could be seen a small amount of gravel, showing the characteristics of traction current. The conglomerate and the underlying strata were mostly bounded by the scouring and filling structure plane, and the erosion cutting phenomenon was common, indicating that the conglomerate was mostly formed in the strong hydrodynamic environment. Its formation was attributed to the braided channel lag deposition (Figs 2.a-b). Additionally, a small amount of graded bedding could also be observed, which was mainly formed by the gradual weakening of water flow after rapid paroxysmal sedimentation events. Sandstone deposits, including gravelly coarse sandstone, coarse sandstone, medium sandstone, and medium fine sandstone, had been formed on the thin layer of conglomerate and glutenite. Among the various sandstone deposits, gravelly coarse sandstone and coarse sandstone were the most prevalent, exhibiting relatively poor sorting and rounding. Among them, gravelly coarse sandstone and coarse sandstone were developed with trough cross bedding, plate cross bedding and massive bedding (Figs 2.c-d), while medium sandstone and fine sandstone were usually developed with small flowing sand ripple bedding and parallel bedding (Figs 2.e-f). From the bottom to the top, there were obvious vertical facies sequence combinations with coarse-to-fine, such as conglomerate, (gravelly) coarse sandstone, medium sandstone and fine sandstone. There were also 4–10 cm low-energy deposits in falling silt layers or gullies with argillaceous bands between sandstone bodies, exhibiting the characteristics of multi-stage superimposed typical braided river channel bar configuration units. On the sandstone bodies, brown, greyish-green, and variegated floodplain muddy sediments were developed, with rare plant carbon debris, indicating an overall arid climate during that period (Figs 2.g-h). Based on the observations of the entire core section, the channel sediments in the Xiashihezi Formation 1 Member primarily consisted of normal cyclic sediments with meter-scale and grain size becoming finer upward, reflecting that the sedimentary water body decreases upward. The riverbed was dominated by coarse-grained sediments in middle-lower part, supplemented by fine-grained sediments such as the upper embankment and overtopping. The multi-stage cutting and overlapping characteristics of the channel bar were obvious, indicating that the river channel was wide, shallow and rapid, without a fixed embankment and easy to divert (Figs 2.i-j). Based on the above

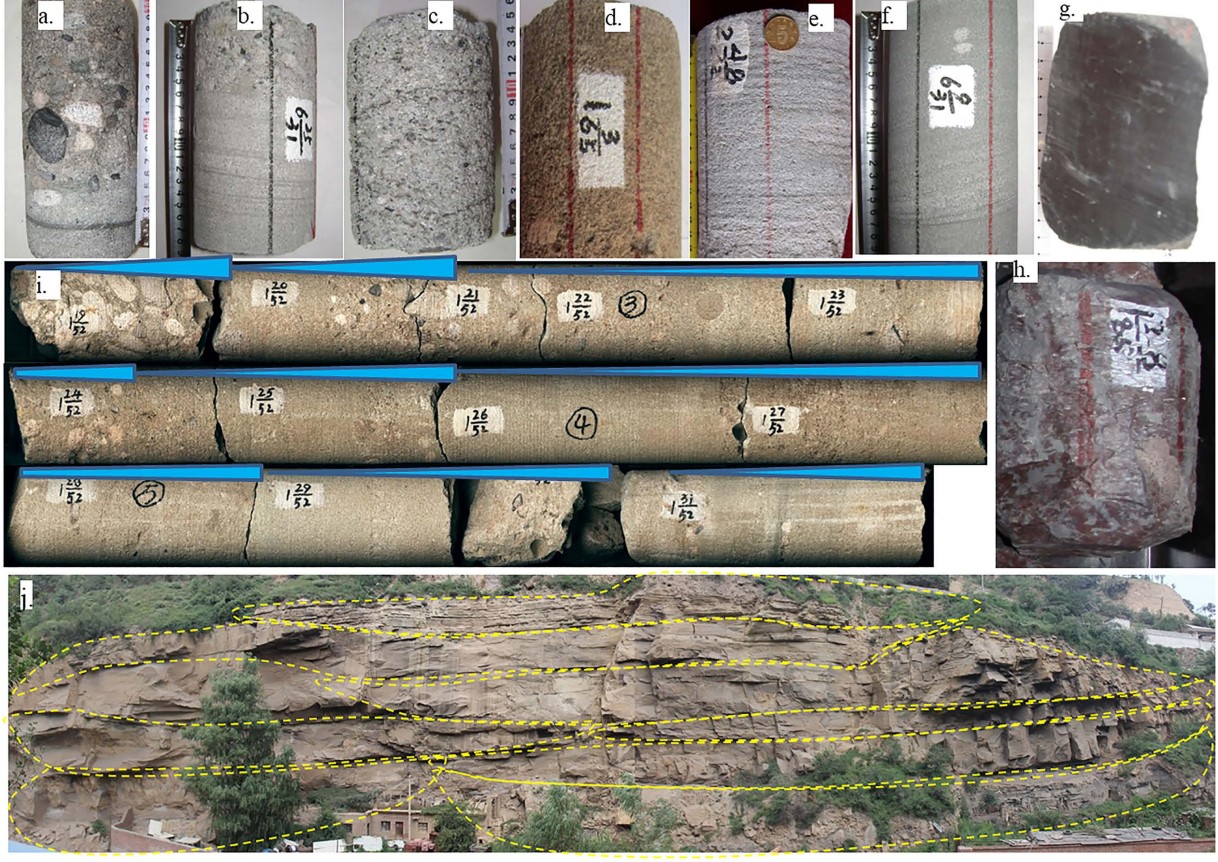

a. channel scouring surface, Jin91, 2959.75m, H1-3; b. channel scouring surface, upper glutenite, lower fine sandstone, Jin91, 3006.6m, H1-1; c. glutenite, trough cross bedding, Jin91, 2985.35m, H1-2; d. massive bedding gravelly coarse sandstone, Jin89, 3081.5m, H1-2; e. parallel bedding medium-fine sandstone, Jin72, 2955.1m, H1-1; f. sandstone with swash bedding, Jin91, 3003.5m, H1-1; g. greyish black mudstone, Jin93, 3025.3m, H1-2; h. Spotted mudstone, Jin69, 2948.7m, H1-2; i. the sandstone bodies in the river channel were stacked in multiple periods, and the characteristics of meter scale sedimentary cycles were obvious., Jin21, H1-1, 2874.95-2877.85m; core diameter is 10cm; j.The multi-stage stacking characteristics of the sandstone body in the channel bar of the He1 Member, the Baolou Gou outcrop profile, Baode county, Shanxi Province.

**Fig 2. Typical sedimentary structures of braided river from Xiashihezi Formation in Shilijiahan west zone.**

analysis of lithofacies characteristics, it could be posited that the Xiashihezi Formation 1 Member in this area represented a relatively typical shallow braided river deposit in the absence of a fixed river channel under relatively dry conditions [16,17].

The above description of core sedimentary characteristics, detailed correlation with logging curves, and lithology and vertical combination characteristics, as well as the amplitude and shape displayed by the corresponding logging curves, indicated that braided river sediments could be subdivided into three microfacies in a single well.

Channel bar microfacies were primarily developed in the lower portion of the sandstone body sedimentary cycle. In the channel bar microfacies, trough cross bedding (gravel-bearing) coarse sandstone facies, tabular cross bedding (gravel-bearing) coarse sandstone facies, and massive bedding (gravel-bearing) coarse sandstone facies were developed, indicating robust hydrodynamic sedimentary conditions. Typical channel scouring surfaces and channel lag gravel deposits could be observed at the base. Due to the relatively thin lag deposits observed in the Xiashihezi Formation 1 Member, it could be challenging to differentiate the logging curve characteristics between the thin lag deposits and the channel bar microfacies. Therefore, this paper didn't further subdivide them. The logging curves corresponding to the

channel bar microfacies could be classified into two types, with high amplitude tooth box type as the main type and a few smooth box types (Fig 3).

The channel sandy filling microfacies were typically developed at the upper or top of the sedimentary cycle of the sandstone body. The lithology was predominantly grayish green medium-fine sandstone, exhibiting parallel bedding and small sand-ripple bedding. The cores revealed the enrichment of bright flake mica, indicating that the hydrodynamic conditions were relatively weak. The logging curve was primarily a combination of medium-high amplitude toothed box and medium amplitude toothed bell (Fig 3).

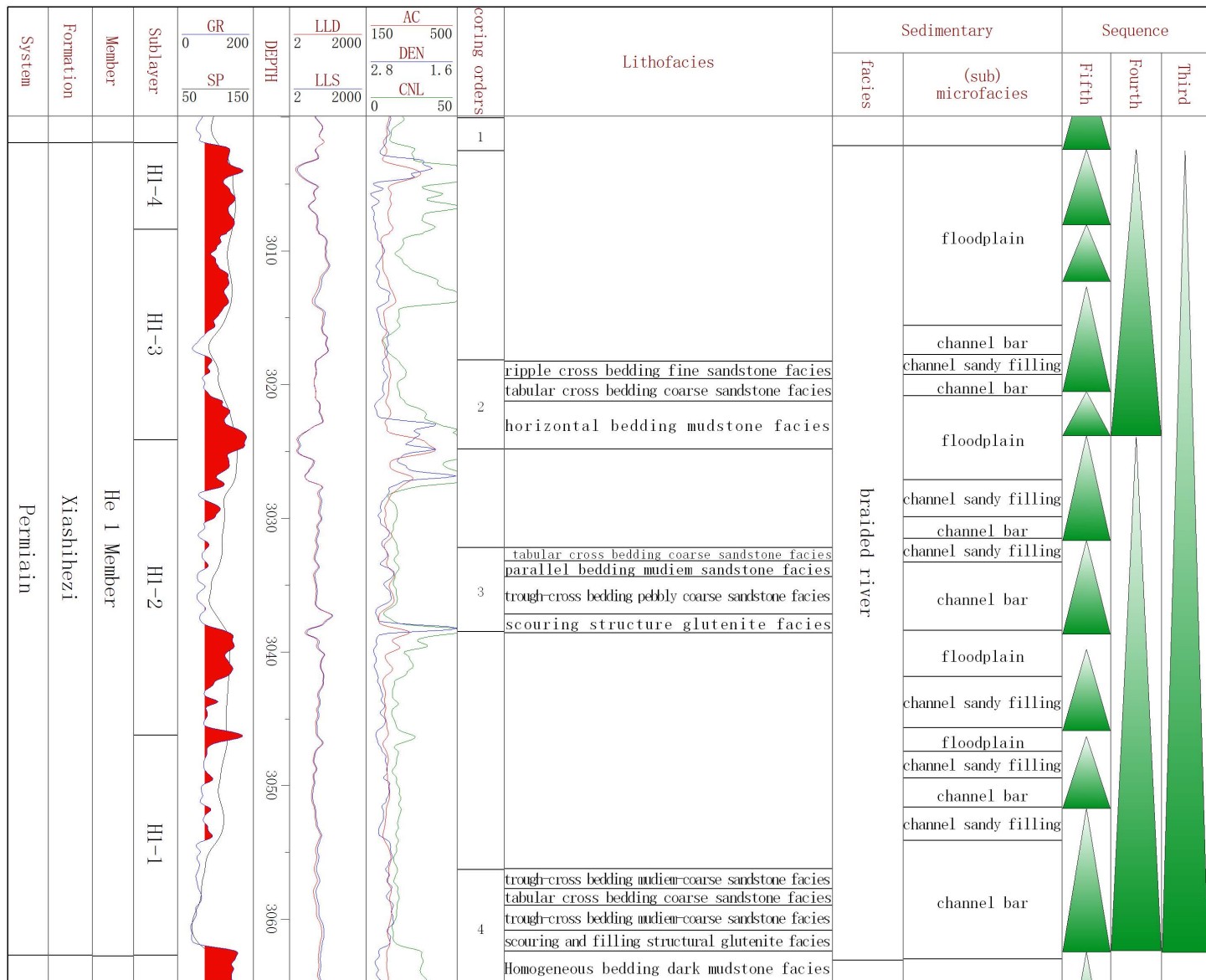

**Fig 3. Lithofacies and sedimentary microfacies division of the Jin93 well from Xiashihezi Formation 1Member.**

Floodplain microfacies: The lithology was predominantly composed of mudstone facies and silty mudstone facies, which were generally developed on the deposition of sandstone body. Occasionally, mudstone drainage structures were observed, and the natural gamma curve (GR) was markedly elevated (Fig 3).

**Description of sedimentary microfacies distribution based on well seismic combination.** The distribution of sedimentary microfacies was determined by analyzing the synthetic seismic records of the drilled wells and the seismic profile of the single well sedimentary microfacies. A comprehensive model of the sedimentary microfacies with logging and seismic response characteristics of the Xiashihezi Formation 1 Member was established (Fig 4), and the vertical and horizontal distribution of sedimentary microfacies was studied [18,19]. The seismic facies response characteristics of the three main sedimentary microfacies of the target layer were as follows.

In the channel bar microfacies, due to the strong acoustic impedance difference between the thick sandstone body and the surrounding rock (mudstone), it mainly presented the short axis reflection of low-frequency strong wave trough (Fig 4 and Fig 5.a). The channel sandy filling microfacies exhibited low-frequency, medium to weak trough reflections. The acoustic impedance difference between the medium-fine sandstone deposited under relatively weak hydrodynamic conditions and the surrounding rock was relatively small, and the reflection coefficient at the reflection interface was relatively small (Fig 4 and Fig 5.a), resulting in relatively medium-weak trough seismic reflection. The floodplain microfacies, with weak amplitude reflection, were primarily low-energy sedimentary assemblages of fine sandstone and mudstone, with thin interbedded deposition (Fig 4 and Fig 5.a). This served as the basis for microfacies correlation, and enabled the lateral thickness change of the sandstone body to be explored by using the value amplitude change of the amplitude attribute. Ultimately, this facilitated the completion of the microfacies analysis of the connection wells correlation section (Fig 5.b).

The detailed calibration results of sedimentary microfacies and seismic profile demonstrated that the greater the thickness of the sandstone, the stronger the trough amplitude. When the sandstone thickness was constant, the greater the proportion of coarse-grained sandstone, and the greater the absolute value of the trough amplitude attribute, that was, the absolute value of the maximum trough amplitude of flood plain, channel filling, and channel bar microfacies, there was a gradually increasing trend. Accordingly, the research findings on the relationship between sandstone body thickness

| Sedimentary facies | | Logging model | | | | | Logging characteristics | Seismic reflection characteristics | Seismic reflection description | well |
|---|---|---|---|---|---|---|---|---|---|---|
| facies | (Sub) microfacies | Sublayer | GR 0–250 / SP 0–100 | DEPTH | LLD 2–2000 / LLS 2–2000 | AC 150–500 / DEN 2.8–1.6 / CNL 0–50 | microfacies | | | | |
| Braided river | Channel bar | H1-3, H1-2 | | 2920 | | | channel bar / channel infilling / channel bar / channel infilling / channel bar | High amplitude box-shaped, with sudden contact at the top and bottom or sudden change at the bottom and gradual change at the top. | | Middle-strong trough amplitude | J72P5 |
| | Channel infilling | H1-3, H1-2 | | 2840 | | | channel infilling | Medium amplitude box-shaped, bell-shaped, with abrupt or gradual contact at the top and bottom. | | Middle-weak trough amplitude | Jin7 |
| | floodplain | H1-1 | | 3080 | | | floodplain | Low amplitude flatness, serration, and anomaly low curve amplitude | | Weak trough amplitude | Jin143 |

**Fig 4. Logging facies and seismic facies characteristics of the Xiashihezi Formation 1 Member.**

and amplitude demonstrated that the channel boundary could be effectively described by the maximum trough amplitude attribute [20,21]. Furthermore, the morphological characteristics of different sedimentary microfacies could be effectively described by employing the corresponding relationship between the maximum trough amplitude range and different microfacies. The results of the above analysis of single well sedimentary microfacies and connected well sedimentary microfacies indicated that the amplitude of weak trough amplitude was primarily associated with floodplain deposits, while the medium trough amplitude was predominantly linked to channel filling microfacies. The most important channel bar microfacies had the characteristics of the strong trough amplitude (Fig 4, Fig 5). The maximum trough amplitude attribute plane for the main target layer was extracted (Fig 6.a), which showed a north-south strip (Fig 6.a). The plane distribution of sedimentary microfacies in the Xiashihezi Formation 1 Member was finally obtained by mutual calibration and constraint between the attribute of maximum trough amplitude and the single well sedimentary microfacies, the connected

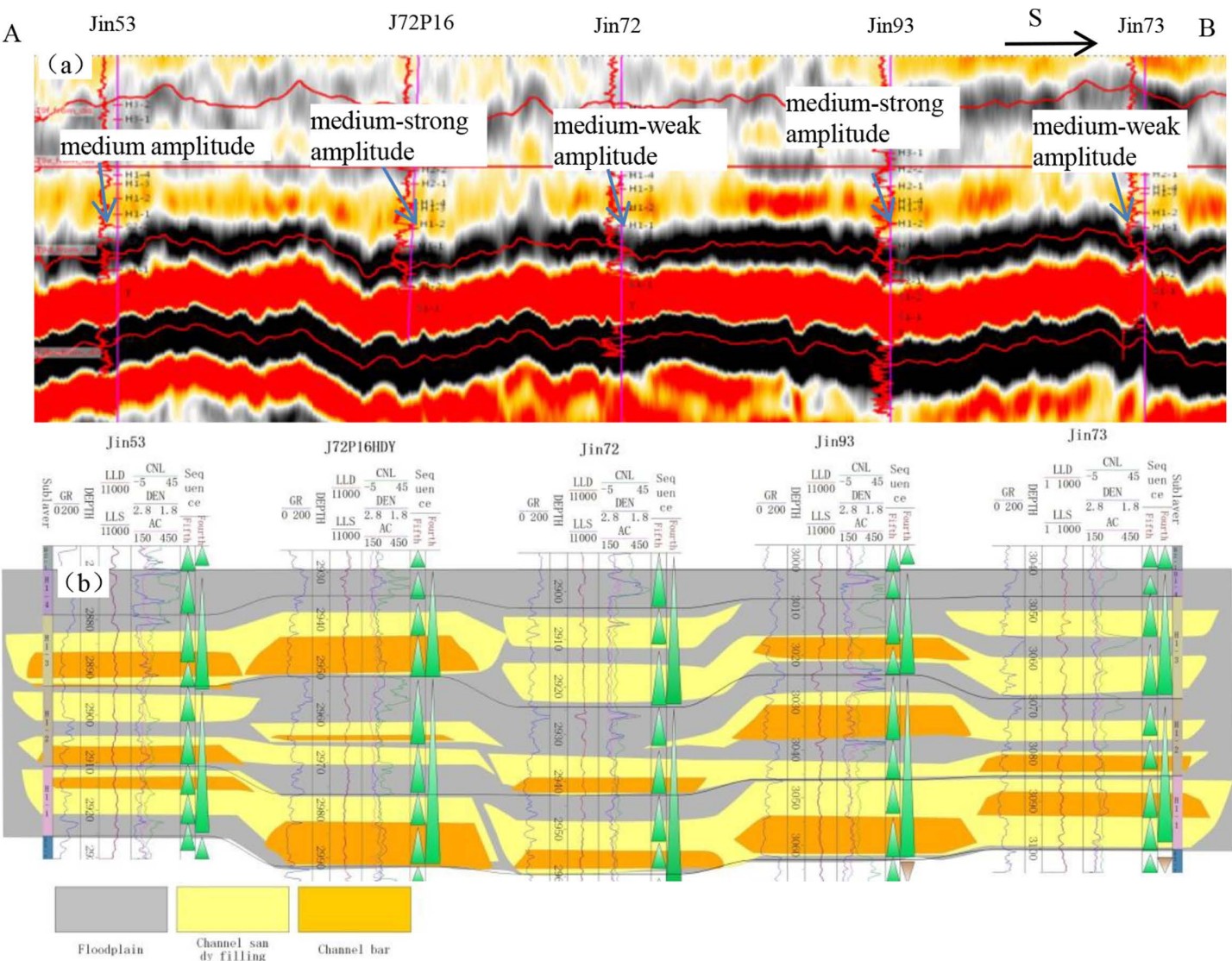

**Fig 5. Seismic profile and sedimentary microfacies of the Xiashihezi Formation 1 Member in the Shilijiahan west zone.**

well sedimentary microfacies [21,22]. The study showed that there were 5 braided channels distributed in the Xiashihezi Formation 1 Member in the Shiligahan west zone, which were named Ch1, Ch2, Ch3, Ch4 and Ch5 respectively from west to East, with a north-south strip distribution. The scale of Ch3, Ch4 and Ch5 channels in the Middle East was larger than that of Ch1 and Ch2 channels in the West (Fig 6.b).

### Reservoir microstructure and pore structure characteristics

**Reservoir microscopic characteristics description.** According to the statistical analysis of thin section data in the study area, the reservoir lithology of Xiashihezi Formation 1 Member in the area was mainly lithic (gravel-containing) coarse sandstone and lithic quartz (gravel-containing) coarse sandstone (Figs 7.a-c), with a small amount of fine glutenite and medium sandstone. The rock clastic was mainly composed of quartz and rock debris, with less feldspar. The lithology in the Xiashihezi Formation 1 Member was predominantly coarse sandstone and gravelly coarse sandstone. In addition to using thin section observation, Bappa Mukherjee et al. (2021) proposed the use of hierarchical clustering analysis (HCA) in lithology identification, which can achieve lithology identification using conventional logging curves. This is also a highly recommended method that can identify lithology throughout the entire wellbore [23]. The average content of quartz in clastic rocks from the Xiashihezi Formation 1 Member was 69.9%, with the particle size primarily comprising coarse grains and gravelly coarse grains. The interstitial materials were primarily composed of argillaceous matrix and authigenic calcite, kaolinite, chlorite, and other cements. The sorting of sandstone particles was predominantly medium to poor, with a sub-angular roundness, followed by sub-circular, and the overall textural maturity was low. Clastic grains were supported with the ways of point line contact, and concave convex contact (Fig 7). The cementation type was mainly pore cementation. From the perspective of the main rock types of the reservoir, it was matched with the sedimentary types with variable braided river dynamic conditions in the region.

The statistical results of porosity and permeability from different types of lithology showed that the physical properties of gravel-bearing coarse sandstone and coarse sandstone lithology reservoirs were significantly better than those of sandy conglomerate, medium sandstone, fine sandstone, and other lithology (Fig 8.a). This suggested that the reservoir development in this area was controlled by sedimentary microfacies. The correlation between porosity and permeability

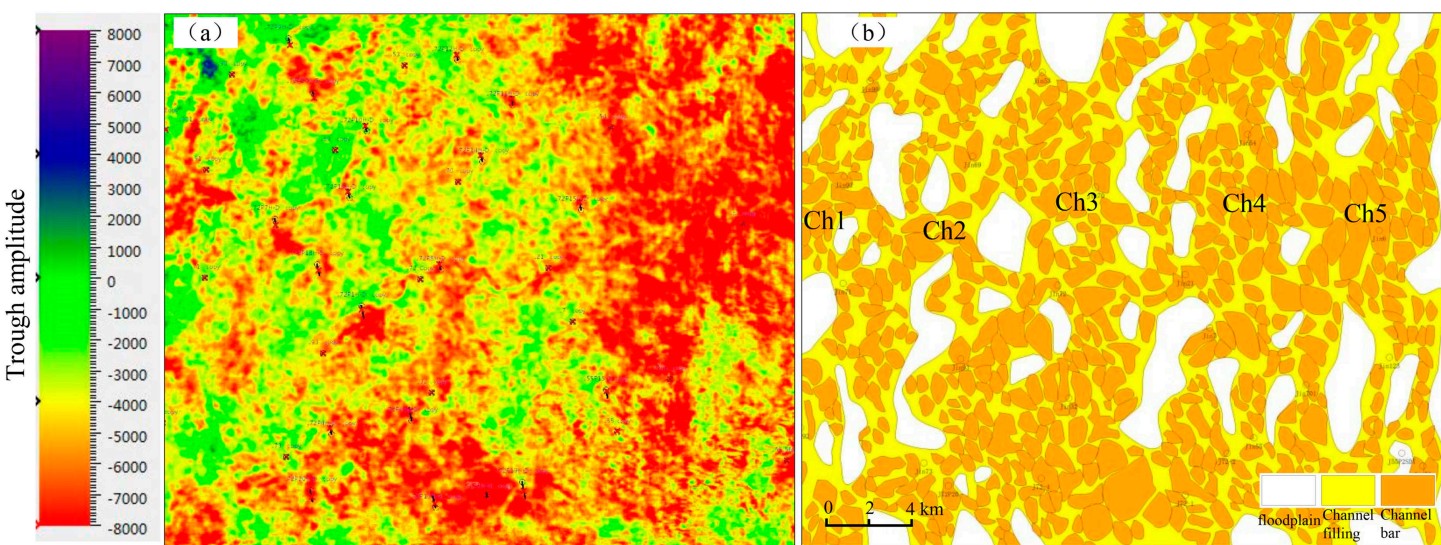

**Fig 6. Seismic attributes and sedimentary facies of the Xiashihezi Formation 1 Member from the Shilijiahan west zone in Hangjinqi area, Ordos basin.**

of samples from the Xiashihezi Formation 1 Member in this area was good, indicating that the reservoir space type is mainly pore type (Fig 8.b). The porosity value ranged from 0.4% to 18.7%, with an average of 7.6% (Fig 9.a). The majority of samples exhibited porosity values below 15%. Permeability values ranged from 0.007mD to 15.1mD, with an average of 0.53mD. The permeability of 87% of the samples was less than 1mD (Fig 9.b). Therefore, the Xiashihezi Formation 1 Member in the study area was generally a low-porosity, ultra-low-porosity, ultra-low-permeability (tight) reservoir.

The experimental results of injecting thin section identification and image analysis indicated that the reservoir space type of the reservoir in the study area was primarily intergranular dissolved pores, intragranular dissolved pores, and primary intergranular pores. Additionally, mold pores and intergranular micropores were also relatively developed, with a

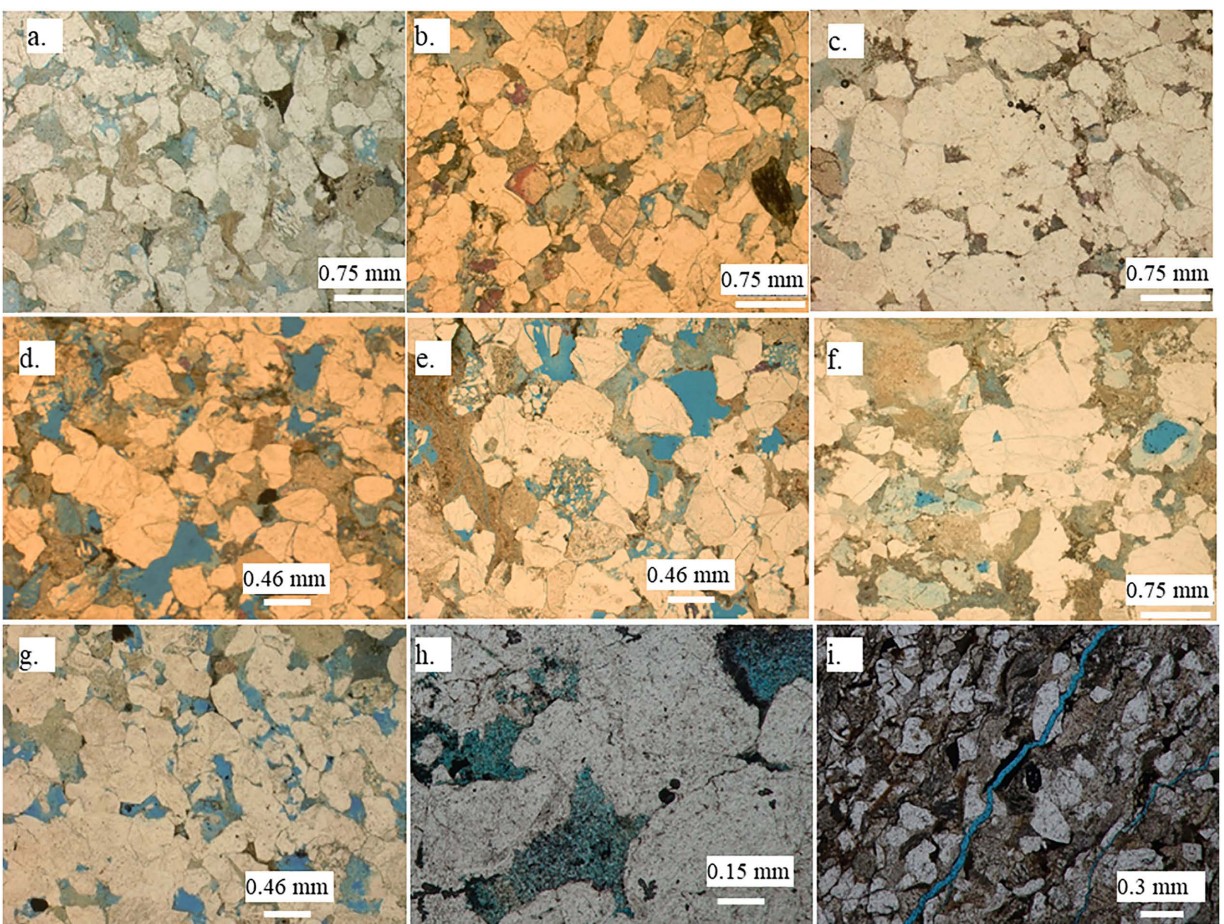

a. medium to coarse-grained lithic sandstone, mainly composed of micropores, distributed in authigenic clay intergranular, mudstone, Jin 72, 2954.1m, H1-1; b. gravel-bearing lithic coarse sandstone, Jin 21, 2834.37m, H1-3; c. lithic quartz coarse sandstone, Jin 89, 3081.97m, H1-2. d. lithic coarse sandstone, with intergranular dissolution pores and intergranular pores developed, clay and some particles containing many micropores, Jin 21, 2841.14m, H1-3; e. lithic coarse sandstone, with intergranular dissolution pores mostly, with a small amount of intergranular dissolution pores, Jin 91, 2991.06m, H1-2; f. lithic medium to coarsem sandstone with occasional distribution of intragranular dissolution pores, Jin 91, 2962.99 m, H1-4. g. lithic quartz coarse sandstone, developed intergranular dissolution pores, intragranular dissolution pores, and intergranular pores, Jin72, 2941.04m, H1-2. h. quartz sandstone with gravel and coarse-grained debris, developed intergranular micropores of kaolinite, Jin 70, 2908.87m, H1-1. i. fine sandstone with muddy, developed microfractures, Jin 71, 3041.74m, H1-1. All are blue cast thin section

**Fig 7. Photos showing pore type from He1 Member gas pool in Jin72 well block, Shilijiahan aera.**

small amount of microfractures. The content of intergranular dissolved pores was the highest in the secondary dissolved pores, followed by intragranular dissolved pores and mold pores. The specific characteristics of pore types, such as dissolution intergranular pores, whose dissolution phenomena were evident at the edges of pores, and some of them were in the shape of dissolution harbors (Fig 7.d-e). Mold pores and intragranular dissolution pores, the majority of these were formed by complete or partial dissolution of feldspar and soluble rock fragments (Fig 7.f). The primary intergranular pores were triangular and multilateral in shape, and the pore edges were mostly straight, without obvious dissolution (Fig 7.g). Intergranular micropores were primarily formed between the grains of bookworm-like kaolinite after the filling of authigenic kaolinite distributed between clastic particles (Fig 7. h). Microfractures were typically developed in relatively tight reservoirs, along grain boundaries or cutting clastic particles (Fig 7.i). A detailed analysis of the reservoir space indicated that intergranular dissolved pores contained some primary pores, which were the result of further dissolution on the basis of primary pores. In summary, the reservoir space type in this area was mainly pore type, including intergranular dissolved pores, intragranular dissolved pores, primary intergranular pores, mold pores, and intergranular micropores, with a small amount of microfractures developed.

**Analysis of reservoir pore structure characteristics.** Pore structure research is a fundamental aspect of reservoir microscopic analysis. In particular, low-permeability tight sandstone reservoirs exhibit distinctive microscopic pore structure characteristics and seepage mechanisms that regulate the migration and accumulation of oil and gas, as well as the productivity of gas wells [24,25]. The size, distribution, uniformity, and other characteristics of pore throats can

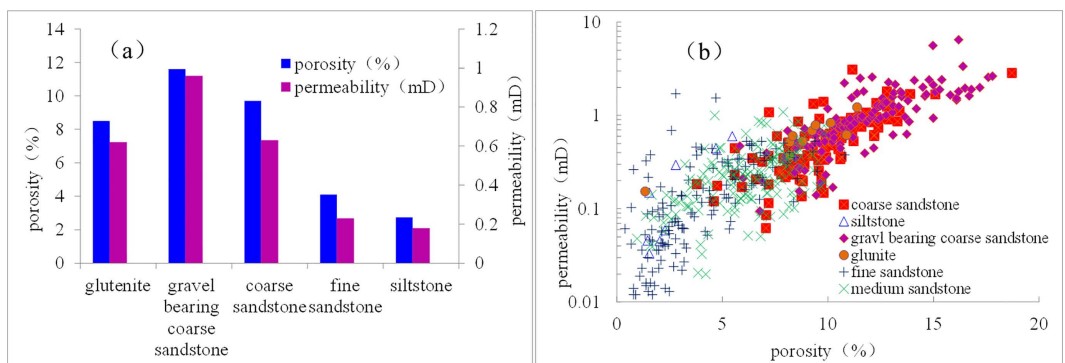

**Fig 8. Distribution characteristics and correlation relationship of porosity and peamibility of different lithology samples from Xiashihezi Formation 1 Member in Shilijiahan west zone.**

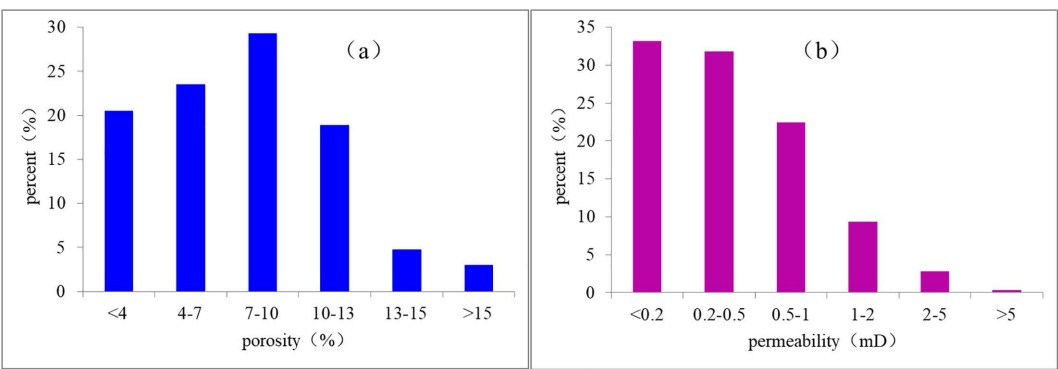

**Fig 9. Distribution characteristics of porosity and peamibility of samples from Xiashihezi Formation 1 Member in Shilijiahan west zone.**

be reflected by the throat radius, skewness, sorting coefficient, and other parameters of the mercury injection capillary pressure curve [24,26,27]. Mercury injection data from a variety of reservoir samples from the Xiashihezi Formation 1 Member of the Shilijiahan west zone were sorted and analyzed. The resulting data were used to calculate the mercury injection parameters (Table 1), which were then used to create a mercury injection curve (Fig 10). This curve was used to study the microscopic pore structure of low-permeability tight sandstone reservoirs.

The inclination of the mercury injection curve to the lower left indicated that the coarser throat skewness is associated with greater reservoir permeability. The wider the mercury injection curve platform is, the better the sorting performance of the throat is. The results of the study indicated that the mercury intrusion curves of the samples in the study area could be divided into four categories (Fig 10), with the corresponding mercury intrusion parameters enumerated in Table 1. The class I curve of capillary pressure was generally concave to the bottom left of the map, with a slight coarse skewness (Fig 10.a). The average displacement pressure was 0.29 MPa, and the average median pressure was 2.42 MPa. The average throat radius was 0.37 µm, and the maximum mercury saturation was 91.5% (Table 1). The class II curve was a steep slope type with medium skewness (Fig 10.a). The capillary pressure curve of the rock samples was steeper than that of the Class I samples, indicating poor sorting. The average displacement pressure was 0.52 MPa, and the average median pressure was 8.27 MPa (Table 1). The average throat radius was 0.16 µm, and the maximum mercury saturation was 88.1%. The Class III capillary pressure curve of the rock samples was slightly smoother than that of the Class II samples, with a slight fine skewness (Fig 10.a). The average displacement pressure was 1.2 MPa, and the average median pressure was 20.2 MPa. The average throat radius was 0.08 µm, and the maximum mercury saturation was 67.6% (Table 1).

**Table 1. Statistical table of reservoir pore throat characteristic parameters in Shilijiahan west zone.**

| Curve type | Lithologic type | Microfacies type | Porosity (%) | Permea-bility (mD) | Displacement pressure (MPa) | Median pressure (MPa) | Average throat radius (µm) | Maximum mercury saturation (%) | Sorting coefficient |
|---|---|---|---|---|---|---|---|---|---|
| class I | gravelly coarse sandstone, coarse sandstone | Channel bar | 13.8 | 1.35 | 0.29 | 2.42 | 0.37 | 91.5 | 0.35 |
| class II | coarse sandstone, gravelly coarse sandstone | Channel bar | 9.3 | 0.91 | 0.52 | 8.27 | 0.16 | 88.1 | 0.39 |
| class III | medium sandstone, medium coarse sandstone | Channel filling | 5.5 | 0.3 | 1.2 | 20.2 | 0.08 | 67.6 | 0.09 |
| class IV | medium fine sandstone, siltstone | Channel filling, floodplain | 3.6 | 0. 15 | 4.6 | / | 0.02 | 41.2 | 0.02 |

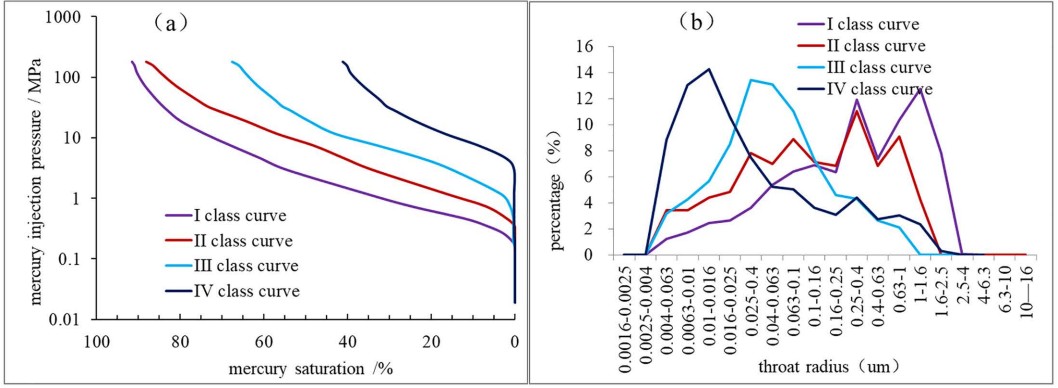

**Fig 10. characteristics of capillary pressure curve and throat radius distribution of sandstone from the Xiashihezi Formation 1 Member in the shilijiahan west zone.**

The skewness of the capillary pressure curve of the class IV type of rock samples was the smallest (Fig 10.a), indicating that the pore throat was very small and generally non-reservoir. The above indicated that the Class II curve of mercury intrusion was not only controlled by the original sedimentation, but also by the dissolution effect of later diagenesis, which led to the generation of uneven pores. Therefore, it was also an important reason for the complex pore structure of the reservoir in this area.

The pore throat distribution standard for clastic rock reservoir evaluation defines the pore throat greater than 1 µm as coarse throat, the pore throat of 0.25 µm~1 µm as middle throat, the pore throat of 0.025 µm~0.25 µm as fine throat, and the pore throat of less than 0.025 µm as micro throat. A mapping of the pore throat radius of the four kinds of rock sample types was presented in Fig 10.b. The characteristics of the pore throat radius distribution curve of the class I of samples were as follows. The curve was typical of a coarse middle throat type, with the majority of samples throat radius distributed at 0.25 µm to 2.5 µm (Fig 10.b). The middle throat and coarse throat are developed. The class II type of sample was the middle throat multi-peak type, with a wide peak. The range of throat radius was mostly at 0.025 µm to 1.6 µm, indicating that the fine throat and middle throat were relatively developed (Fig 10.b). The class III type of samples was medium fine throat single peak type, and the throat radius of the class III type of samples ranged from 0.01 µm to 0.25 µm. The IV type of samples was a fine-micro throat single-peak type. The throat radius distribution was mostly between 0.004 µm and 0.4 µm (Fig 10.b), exhibiting clear micro throat characteristics and a small amount of fine throat.

A review of the pore and throat characteristics in the study area, as analyzed in the context of reservoir space types, revealed that the pore sizes were primarily medium and small (micro) pores, with the presence of locally developed large pores. The throat radius, as reflected by mercury injection curves of different types of reservoirs, indicated that the throat as a whole was dominated by medium and fine throats (Fig 11.a). The pore throat combination included predominantly medium pore with medium throat type and medium-small pore with fine throat type (Fig 11.b), which was relatively poor.

**Effect of sedimentation on reservoir development.** The sedimentary environment and facies exerted a profound influence on the formation and distribution of reservoirs. The water dynamic conditions of sedimentary in clastic rocks in different facies zones were markedly distinct. There were notable differences in mineral composition, grain size, and interstitial materials of reservoir rocks, which profoundly impacted the physical properties, pore structure, and percolation characteristics of sandstone reservoirs, resulting in significant alterations in reservoir performance in different facies zones [28–31]. Consequently, the sedimentary facies represented a pivotal factor in controlling reservoir performance.

The distribution characteristics of porosity and permeability of different samples in the study area (see Fig 8.a) were closely related to the sedimentary microfacies. The average porosity of gravelly coarse sandstone and coarse sandstone in braided river channel bars was 10.8%, with a permeability of 0.79 mD. The average porosity of sandstone and fine sandstone in channel filling was 5.45%, with a permeability of 0.34 mD. The porosity of siltstone in floodplain was 3%,

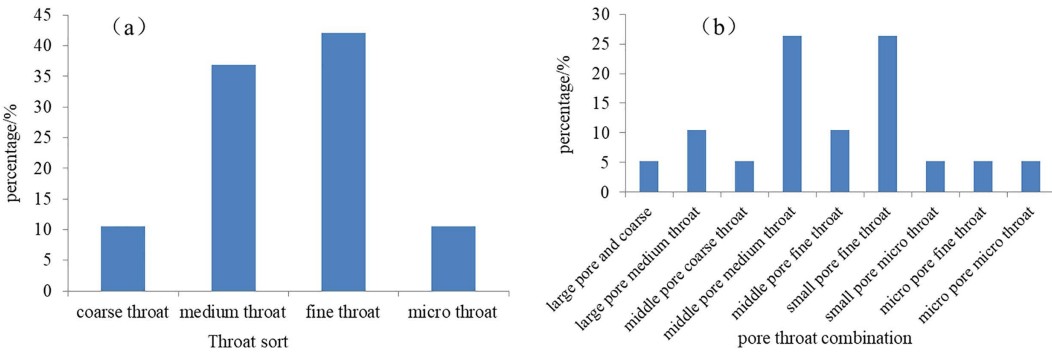

**Fig 11. Percentage of different levels pore throats and pore combination Xiashihezi Formation 1 Member in Shilijiahan west zone.**

with a permeability of 0.2 mD. The physical properties of braided river channel bar were the most favorable, while those of channel filling sandstone and floodplain sandstone were relatively poor (Fig 8). The braided river channel bar, subject to a strong hydrodynamic effect, was scoured by high-energy, stable water flow, resulting in the formation of a thick sandstone body, a large sandstone grain size, a small content of rock debris and matrix, and a relatively good sorting. Furthermore, it was susceptible to dissolution during diagenesis [32,33], which resulted in the development of intergranular and intra-granular dissolved pores (Fig 7.d-f). This process facilitated the formation of sandstone reservoirs with desirable physical properties. Channel filling and floodplain sandstones were relatively fine in grain size and thin in thickness, rendering them more susceptible to the strong influence of the content of argillaceous and carbonate cements (Fig 2.e-f). The content of plastic particles and matrix, such as argillaceous rock debris and mica, in the reservoir was high (Fig 2.e-f, Fig 7.i), resulting in poor reservoir physical properties. It was evident that the braided river channel bar microfacies represented a favorable reservoir microfacies for exploration and development.

The aforementioned research demonstrated that the physical properties of sandstone bodies in diffirent sedimentary microfacies were distinct. However, even within the same sedimentary microfacies, the changes of sediment composition with hydrodynamic conditions would also be different, resulting in the observed differences in physical properties [32]. In the central and eastern part of the study area, the river channels designated Ch3, Ch4 and Ch5 exhibited a pronounced hydrodynamic force, low clay matrix content, an advanced pattern of sandstone body inheritance, a considerable width and thickness of the river channel, the presence of coarse sandstone particles, a higher original porosity and permeability. These conditions were also conducive to the occurrence of dissolution during the diagenetic stage, ultimately forming high-quality reservoirs with favourable physical properties, particularly in Jin32 well block at the intersection of multiple rivers in the south of the Ch3 river channel. The braided river channel was characterized by a wide, expansive channel bar, which was densely developed and conducive to reservoir development (Fig 6.b). While the sandstone bodies in the channel bar of the Ch1 and Ch2 river channels in the west were situated in relatively weaker hydrodynamic conditions than those in the middle and east, the river channel size and the grain size of sedimentary rocks were smaller than those in the Middle East, so the final reservoir physical properties were poor (Fig 6.b).

The microscopic pore structure of reservoir rock exerted a significant influence on the fluid reservoir and percolation capacity [34], which represented a pivotal factor in the evaluation of reservoir effectiveness. The porosity of braided river channel bar sandstone reservoirs was predominantly greater than 9% (Fig 8.a), with a maximum mercury saturation of over 80% and a displacement pressure of less than 0.6 MPa. Additionally, the average throat radius was typically greater than 0.16 µm, indicating that the pore throat structure was relatively good. Although the braided river channel bar sandstones were poorly sorted relative to other microfacies sandstones, the reservoir heterogeneity was enhanced due to the development of dissolution (Fig 7.d-f). Consequently, the number of large throats also increased, reflecting that the reservoir percolation ability was enhanced and the physical properties were improved. Although the channel filling sandstones were relatively well sorted, the majority of the sediments were medium sandstones with a small grain size, which were prone to form small pores and fine throats. These included residual intergranular pores, feldspar intragranular dissolved pores, and intergranular filling kaolinite intergranular micropores (Fig 7.g-i). Consequently, the reservoir pore structure was poor.

## Conclusions

(1) Based on an analysis of the lithology of cores, sedimentary facies marks, vertical sedimentary variation characteristics, sandstone body combination morphology and three-dimensional distribution, it can be concluded that the sedimentary type of the Xiashihezi Formation 1 Member in the Shiligahan west zone belongs to the wandering shallow braided river category. This is evidenced by the presence of approximately five braided river channels developed on

a north-south long strip distribution plane. The river channels in the middle and east are larger than those in the west, and the channel bar is the best favorable sedimentary microfacies for reservoir development. An effective method for plane analysis of sedimentary microfacies is to establish the relationship among lithofacies, logging facies and seismic facies, and to select geophysical attributes with high correlation with sedimentary microfacies to constrain the characterisation of braided channel and its internal microfacies boundary.

(2) The primary reservoir rock of the Xiashihezi Formation 1 Member is gravelly coarse sandstone, and the reservoir space types are primarily intergranular dissolved pores and intragranular dissolved pores, predominantly medium pores and small pores. The throat radius as reflected by mercury injection curves of varying reservoir types demonstrates the throat is dominated by medium and fine throats. The pore throat combination is of the medium pore medium throat type and the medium to small pore fine throat type. The seepage ability is medium to poor. This is a typical ultra-low porosity, low porosity, low permeability, tight porous reservoir. The sedimentary microfacies and their hydrodynamic conditions are of paramount importance in controlling the development of the Xiashihezi Formation 1 Member reservoir, as well as the reservoir thickness, lithologic grain size and filling material composition. This, in turn, gives rise to differences in reservoir macro distribution, reservoir physical properties and pore structure. This understanding provides a basis for facies-controlled reservoir geophysical prediction.

## Supporting information

**S1 Fig 8. Distribution characteristics and correlation relationship analysis data of porosity and peamibility analysis data of different lithology samples from Xiashihezi Formation 1 Member in Shilijiahan west zone.**
(XLSX)

**S2 Fig 9. Distribution characteristics analysis data of porosity and peamibility of samples from Xiashihezi Formation 1 Member in Shilijiahan west zone.**
(XLSX)

**S3 Fig 10. Capillary pressure curve and throat radius distribution analysis data of sandstone from the Xiashihezi Formation 1 Member in the shilijiahan west zone.**
(XLSX)

**S4 Fig 11. Different levels pore throats and pore combination analysis data from Xiashihezi Formation 1 Member in Shilijiahan west zone.**
(XLSX)

## Author contributions

**Writing – original draft:** Hongtao Li, Tao Liu.

**Writing – review & editing:** Qingbin Liu.

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
