## [Decision Letter · Decision Letter 0]

2 Dec 2024

PONE-D-24-37259Sedimentary and reservoir characterization of wandering braided river: a case of the Xiashihezi Formation 1 Member in Shilijiakhan west zone, Hangjinqi, Northern Ordos BasinPLOS ONE

Dear Dr. Li,

Thank you for submitting your manuscript to PLOS ONE. After careful consideration, we feel that it has merit but does not fully meet PLOS ONE’s publication criteria as it currently stands. Therefore, we invite you to submit a revised version of the manuscript that addresses the points raised during the review process.

We have now received two reports on the manuscript "Sedimentary and reservoir characterization of wandering braided river: a case of the Xiashihezi Formation 1 Member in Shilijiakhan west zone, Hangjinqi, Northern Ordos Basin" by Hongtao Li et alii, which was submitted to Plos One. Based on these reports, minor revision is required before the manuscript can be reconsidered for publication. Depending on the revision, the manuscript may need to be re-reviewed. Both referees suggest ways in which the paper and figures need to be revised, and one of them suggests that the English needs a careful checking. I agree with this and warm the authors that if they resubmit a revised version with poor quality English, the article will be automatically rejected. Please make sure also that you deal with the scientific criticisms and try to make the article a little bit more accessible to the general reader and to a broader audience (add new international references and expand the discussion by comparing the case study with others elsewhere).

We look forward to receiving your revised manuscript.

Kind regards,

Vittorio Scisciani

Academic Editor

PLOS ONE

Journal Requirements:

3. Thank you for stating the following financial disclosure: “Fund projects: "Research on the Development Potential of Dense and High Water-bearing Gas Reservoirs in Dongsheng Gas Field" (P23133) by the Ministry of Science and Technology of Sinopec and National Major Special Project "Development of Large Oil and Gas Fields and Coalbed Methane" (No. 2016ZX05002-006)”

4. We note that your Data Availability Statement is currently as follows: “All relevant data are within the manuscript and in Supporting Information files.”

Please confirm at this time whether or not your submission contains all raw data required to replicate the results of your study. Authors must share the “minimal data set” for their submission. PLOS defines the minimal data set to consist of the data required to replicate all study findings reported in the article, as well as related metadata and methods (https://journals.plos.org/plosone/s/data-availability#loc-minimal-data-set-definition). For example, authors should submit the following data: - The values behind the means, standard deviations and other measures reported; - The values used to build graphs; - The points extracted from images for analysis. Authors do not need to submit their entire data set if only a portion of the data was used in the reported study. If your submission does not contain these data, please either upload them as Supporting Information files or deposit them to a stable, public repository and provide us with the relevant URLs, DOIs, or accession numbers. For a list of recommended repositories, please see https://journals.plos.org/plosone/s/recommended-repositories. If there are ethical or legal restrictions on sharing a de-identified data set, please explain them in detail (e.g., data contain potentially sensitive information, data are owned by a third-party organization, etc.) and who has imposed them (e.g., an ethics committee). Please also provide contact information for a data access committee, ethics committee, or other institutional body to which data requests may be sent. If data are owned by a third party, please indicate how others may request data access.

5. We note that Figure 1 in your submission contain map/satellite images which may be copyrighted. All PLOS content is published under the Creative Commons Attribution License (CC BY 4.0), which means that the manuscript, images, and Supporting Information files will be freely available online, and any third party is permitted to access, download, copy, distribute, and use these materials in any way, even commercially, with proper attribution. For these reasons, we cannot publish previously copyrighted maps or satellite images created using proprietary data, such as Google software (Google Maps, Street View, and Earth). For more information, see our copyright guidelines: http://journals.plos.org/plosone/s/licenses-and-copyright. We require you to either (a) present written permission from the copyright holder to publish these figures specifically under the CC BY 4.0 license, or (b) remove the figures from your submission:

a. You may seek permission from the original copyright holder of Figure 1 to publish the content specifically under the CC BY 4.0 license. We recommend that you contact the original copyright holder with the Content Permission Form (http://journals.plos.org/plosone/s/file?id=7c09/content-permission-form.pdf) and the following text: “I request permission for the open-access journal PLOS ONE to publish XXX under the Creative Commons Attribution License (CCAL) CC BY 4.0 (http://creativecommons.org/licenses/by/4.0/). Please be aware that this license allows unrestricted use and distribution, even commercially, by third parties. Please reply and provide explicit written permission to publish XXX under a CC BY license and complete the attached form.” Please upload the completed Content Permission Form or other proof of granted permissions as an "Other" file with your submission. In the figure caption of the copyrighted figure, please include the following text: “Reprinted from [ref] under a CC BY license, with permission from [name of publisher], original copyright [original copyright year].”

b. If you are unable to obtain permission from the original copyright holder to publish these figures under the CC BY 4.0 license or if the copyright holder’s requirements are incompatible with the CC BY 4.0 license, please either i) remove the figure or ii) supply a replacement figure that complies with the CC BY 4.0 license. Please check copyright information on all replacement figures and update the figure caption with source information. If applicable, please specify in the figure caption text when a figure is similar but not identical to the original image and is therefore for illustrative purposes only. The following resources for replacing copyrighted map figures may be helpful: USGS National Map Viewer (public domain): http://viewer.nationalmap.gov/viewer/ The Gateway to Astronaut Photography of Earth (public domain): http://eol.jsc.nasa.gov/sseop/clickmap/ Maps at the CIA (public domain): https://www.cia.gov/library/publications/the-world-factbook/index.html and https://www.cia.gov/library/publications/cia-maps-publications/index.html NASA Earth Observatory (public domain): http://earthobservatory.nasa.gov/ Landsat: http://landsat.visibleearth.nasa.gov/ USGS EROS (Earth Resources Observatory and Science (EROS) Center) (public domain): http://eros.usgs.gov/# Natural Earth (public domain): http://www.naturalearthdata.com/

Additional Editor Comments:

Specific comments by reviewers are listed below:

Reviewer 1

1. The grammar/English writing of the manuscript can be further improved.

2. The geographic location and other information description of the area should be added in the "geological background" section.

3. In Section 2.2, "The plane distribution of sedimentary microfacies in the Xiashihezi Formation 1 Member was finally obtained by mutual calibration and constraint between the attribute of maximum trough amplitude and the single well sedimentary microfacies, the connected well sedimentary microfacies [21]." And " These conditions were also conducive to the occurrence of dissolution during the diagenetic stage, ultimately forming high-quality reservoirs with favourable physical properties”, Additional basis is needed, such as more references.

4. The legend text in Figure 1 of the paper needs to be adjusted, some fonts need to be enlarged, and some layouts are not standardized enough.

5.The amplitude attribute in figure 6.a of the paper lacks color legend.

6. Only layer and sublayer information appears in Figure 3. It is recommended to complete the complete System, Formation and other information.

7. The corresponding well number can be marked in the sedimentary microfacies type, logging mode and seismic response characteristics in Figure 4.

8.The reference format is not standardized and consistent. Please revise carefully according to the requirements of the journal.

Reviewer 2

1. In the part of “Regional geological background”, the last part of “Taiyuan Formation and Shanxi formation were the main source rock series, While Shanxi Formation and Xiashihezi formation were important reservoir rock series, and Shangshihezi formation and Shiqianfeng Formation were regional caprocks, which together formed the upper Paleozoic oil system of lower generation, upper reservoir and top sealing [17]” can be deleted because it has no great connection with the sedimentation and reservoir discussed in the article.

2. The article discusses that "based on the above analysis of lithofacies characteristics, it is considered that the HE1 Member in this area is a relatively typical wandering shallow braided river deposit without fixed river channel under partial drought conditions", and it is necessary to add necessary evidence and references to support the viewpoint of the article.

3. Fig. 2 is mainly based on core observation photos. It is not clear how to extend and change laterally with the view of a hole. It is suggested to add field outcrop photos and elaborate on the vertical and horizontal overlap characteristics of sedimentary microfacies in detail based on these photos.

4. Some of the references are relatively old, so a small number of older documents can be appropriately deleted, and references of nearly three years can be added to make the article more convincing.

Reviewers' comments:

Reviewer's Responses to Questions

**Comments to the Author**

1. Is the manuscript technically sound, and do the data support the conclusions?

Reviewer #1: Yes

Reviewer #2: Yes

2. Has the statistical analysis been performed appropriately and rigorously? 

Reviewer #1: Yes

Reviewer #2: Yes

3. Have the authors made all data underlying the findings in their manuscript fully available?

Reviewer #1: Yes

Reviewer #2: Yes

4. Is the manuscript presented in an intelligible fashion and written in standard English?

Reviewer #1: Yes

Reviewer #2: Yes

5. Review Comments to the Author

Reviewer #1: This paper presents a study of the sedimentary microfacies of the Xiashihezi Formation 1 Member in the Shiligahan area, situated in the northern Ordos Basin. By using core, logging, and 3D seismic data, the sedimentary environment and sedimentary microfacies were successfully identified. The study area was thus determined to be a wandering, shallow braided river sedimentary system. Through calibration and verification among lithofacies, logging facies, and seismic facies, a comprehensive characterization of the plane distribution of braided channels and their internal microfacies in the Xiashihezi Formation 1 Member was achieved. Further analysis of core observation, thin sections, physical properties, and mercury injection curves, particularly through the study and evaluation of pore structure, identified the microscopic characteristics of the reservoir. By integrating macro- and micro-level perspectives, this study elucidates the influence of sedimentation on reservoir development. The article's theme is evident, its structure is logical, and its findings are both novel and practically applicable. After minor revisions, it can be published.

Reviewer #2: This manuscript analyzes the sedimentary characteristics of shallow braided river in He 1 Member of northern Ordos Basin by using the integrated research of geology, logging and seismic, and summarizes the response modes of logging facies and seismic facies of different sedimentary microfacies. Through the research idea of single well sedimentary microfacies analysis, sedimentary connected wells microfacies correlation and plane sedimentary microfacies , combining the sedimentary microfacies from wells and seismic facies response modes, the sedimentary microfacies boundary is accurately delineated, which provides the basis for the macro distribution analysis of reservoir. The microscopic characteristics of reservoir lithology, physical property, reservoir space type and pore structure characteristics are analyzed in detail. Through the combination of macro and micro, the controlling factors of sedimentation on reservoir development are analyzed. The paper has rigorous structure, strong logic and sufficient evidence, and has certain innovation and application value, which provides technical support for further fine description and selection evaluation of gas reservoirs.

I suggest that this article can be published, but there are also some problems that need to be further modified and improved:

1. In the part of “Regional geological background”, the last part of “Taiyuan Formation and Shanxi formation were the main source rock series, While Shanxi Formation and Xiashihezi formation were important reservoir rock series, and Shangshihezi formation and Shiqianfeng Formation were regional caprocks, which together formed the upper Paleozoic oil system of lower generation, upper reservoir and top sealing [17]” can be deleted because it has no great connection with the sedimentation and reservoir discussed in the article.

2. The article discusses that "based on the above analysis of lithofacies characteristics, it is considered that the HE1 Member in this area is a relatively typical wandering shallow braided river deposit without fixed river channel under partial drought conditions", and it is necessary to add necessary evidence and references to support the viewpoint of the article.

3. Fig. 2 is mainly based on core observation photos. It is not clear how to extend and change laterally with the view of a hole. It is suggested to add field outcrop photos and elaborate on the vertical and horizontal overlap characteristics of sedimentary microfacies in detail based on these photos.

4. Some of the references are relatively old, so a small number of older documents can be appropriately deleted, and references of nearly three years can be added to make the article more convincing.

6. PLOS authors have the option to publish the peer review history of their article (what does this mean? ). If published, this will include your full peer review and any attached files.

**Do you want your identity to be public for this peer review?** For information about this choice, including consent withdrawal, please see our Privacy Policy .

Reviewer #1: **Yes: ** Wang Jian

Reviewer #2: No

---

## [Author Response · Author response to Decision Letter 1]

20 Jan 2025

2�Reply to reviewer 1

1. The grammar/English writing of the manuscript can be further improved.

Reply: The reviewer's comments are very good, and the author has made some modifications to the article to avoid grammar and language errors as much as possible. However, considering the author's relatively limited English proficiency, please the expert reviewers forgive us.

2. The geographic location and other information description of the area should be added in the "geological background" section.

Reply: According to the suggestions of the review experts, the geographical location and other information description of the area have been added in the "Geological Background" section.

3. In Section 2.2, "The plane distribution of sedimentary microfacies in the Xiashihezi Formation 1 Member was finally obtained by mutual calibration and constraint between the attribute of maximum trough amplitude and the single well sedimentary microfacies, the connected well sedimentary microfacies [21]." And " These conditions were also conducive to the occurrence of dissolution during the diagenetic stage, ultimately forming high-quality reservoirs with favourable physical properties”, Additional basis is needed, such as more references.

Reply: According to the suggestions of the reviewing experts, references have been added in both sections to support the viewpoint of the article.

4. The legend text in Figure 1 of the paper needs to be adjusted, some fonts need to be enlarged, and some layouts are not standardized enough.

Reply: According to the suggestions of the review experts, the legend text in Figure 1 has been adjusted, and some fonts have been enlarged. Please refer to the revised draft for details.

5.The amplitude attribute in figure 6.a of the paper lacks color legend.

Reply: According to the suggestions of the review experts, a color legend for the amplitude attribute in Figure 6.a has been added.

6. Only layer and sublayer information appears in Figure 3. It is recommended to complete the complete System, Formation and other information.

Reply: According to the suggestions of the review experts, information such as Formation and Member has been added to Figure 3.

7. The corresponding well number can be marked in the sedimentary microfacies type, logging mode and seismic response characteristics in Figure 4.

Reply: According to the suggestions of the review experts, corresponding well numbers have been added to Figure 4.

8.The reference format is not standardized and consistent. Please revise carefully according to the requirements of the journal.

Reply: According to the suggestions of the reviewing experts, the reference format has been carefully revised according to the requirements of the journal.

(3) Response to reviewer 2:

1. In the part of “Regional geological background”, the last part of “Taiyuan Formation and Shanxi formation were the main source rock series, While Shanxi Formation and Xiashihezi formation were important reservoir rock series, and Shangshihezi formation and Shiqianfeng Formation were regional caprocks, which together formed the upper Paleozoic oil system of lower generation, upper reservoir and top sealing [17]” can be deleted because it has no great connection with the sedimentation and reservoir discussed in the article.

Reply: According to expert advice, "Taiyuan Formation and Shanxi Formation were the main source rock series, while Shanxi Formation and Xiashihezi Formation were important reservoir rock series, and Shangshihezi Formation and Shiqianfeng Formation were regional capstones, which together formed the upper Paleozoic oil system of lower generation, upper reservoir and top sealing [17]" has been deleted.

2. The article discusses that "based on the above analysis of lithofacies characteristics, it is considered that the HE1 Member in this area is a relatively typical wandering shallow braided river deposit without fixed river channel under partial drought conditions", and it is necessary to add necessary evidence and references to support the viewpoint of the article.

Reply: According to expert advice, two additional references have been added to support the viewpoints and understanding of this paper.

3. Fig. 2 is mainly based on core observation photos. It is not clear how to extend and change laterally with the view of a hole. It is suggested to add field outcrop photos and elaborate on the vertical and horizontal overlap characteristics of sedimentary microfacies in detail based on these photos.

Reply: According to expert advice, field outcrop photos have been added to further elucidate the vertical and horizontal stacking characteristics of sedimentary microfacies.

4. Some of the references are relatively old, so a small number of older documents can be appropriately deleted, and references of nearly three years can be added to make the article more convincing.

Reply: According to expert advice, 4 relatively old references have been deleted and 4 relatively new references have been added.

Finally, thank you again for the suggestions from the experts in your editorial department. The author has made every effort to revise and improve the original manuscript. If there are any shortcomings, we kindly ask the reviewers to criticize, correct and forgive us!

---

## [Editor Report · Decision Letter 1]

5 Mar 2025

PONE-D-24-37259R1Sedimentary and reservoir characterization of wandering braided river: a case of the Xiashihezi Formation 1 Member in Shilijiakhan west zone, Hangjinqi, Northern Ordos BasinPLOS ONE

Dear Dr. Li,

Thank you for submitting your manuscript to PLOS ONE. After careful consideration, we feel that it has merit but does not fully meet PLOS ONE’s publication criteria as it currently stands. Therefore, we invite you to submit a revised version of the manuscript that addresses the points raised during the review process.

We look forward to receiving your revised manuscript.

Kind regards,

Bappa Mukherjee

Academic Editor

PLOS ONE

Journal Requirements:

Additional Editor Comments:

Editorial comments to improve the quality and presentation:

1. Please incorporate the range of the petrophysical parameters in the abstract.

2. The sentences presented in the abstract is too long, break them and make a simple representation.

3. Please add suitable keywords.

4. In Fig. 6a please mention the name of the parameter presented with a color scale.

5. Please check the axis title of Fig. 10a.

6. If possible, kindly add a dendrogram analysis based on the hierarchical cluster analysis. You may follow the article "Mukherjee, B., Sain, K. Vertical lithological proxy using statistical and artificial intelligence approach: a case study from Krishna-Godavari Basin, offshore India. Mar Geophys Res 42, 3 (2021). https://doi.org/10.1007/s11001-020-09424-8." Otherwise in the discussion section you can add a few lines and make a recommendation for future researchers to consider this analysis.

---

## [Author Response · Author response to Decision Letter 2]

21 Jul 2025

Journal Requirements:

Reply: the references are complete and correct, and an additional reference has been added.

Additional Editor Comments:

Editorial comments to improve the quality and presentation:

1. Please incorporate the range of the petrophysical parameters in the abstract.

Reply According to the requirements, add petrophysical parameters such as reservoir porosity and average permeability in the abstract.

2. The sentences presented in the abstract is too long, break them and make a simple representation.

Reply Following the suggestion, the long sentence has been revised to be as short as possible.

3. Please add suitable keywords.

Reply Added 6 keywords.

4. In Fig. 6a please mention the name of the parameter presented with a color.

Reply In Figure 6.a, the names of geophysical attributes have been added to the color scale.

5. Please check the axis title of Fig. 10a.

Reply Modified the vertical axis title in Figure 10a

6. If possible, kindly add a dendrogram analysis based on the hierarchical cluster analysis. You may follow the article "Mukherjee, B., Sain, K. Vertical lithological proxy using statistical and artificial intelligence approach: a case study from Krishna-Godavari Basin, offshore India. Mar Geophys Res 42, 3 (2021). https://doi.org/10.1007/s11001-020-09424-8." Otherwise in the discussion section you can add a few lines and make a recommendation for future researchers to consider this analysis.

Reply In the discussion section, we add a few lines and make a recommendation for future researchers to consider hierarchical clustering analysis (HCA) analysis. an additional reference of Mukherjee has been added.

---

## [Editor Report · Decision Letter 2]

24 Jul 2025

Sedimentary and reservoir characterization of wandering braided river: a case of the Xiashihezi Formation 1 Member in Shilijiakhan west zone, Hangjinqi, Northern Ordos Basin

PONE-D-24-37259R2

Dear Dr. Li,

We’re pleased to inform you that your manuscript has been judged scientifically suitable for publication and will be formally accepted for publication once it meets all outstanding technical requirements.

Kind regards,

Bappa Mukherjee

Academic Editor

PLOS ONE
---

## [Editor Report · Acceptance letter]

PONE-D-24-37259R2

PLOS ONE

Dear Dr. Li,

I'm pleased to inform you that your manuscript has been deemed suitable for publication in PLOS ONE. Congratulations! Your manuscript is now being handed over to our production team.

Kind regards,

on behalf of

Dr. Bappa Mukherjee

Academic Editor

PLOS ONE